

# Teacher preparedness regarding autism spectrum disorder in the Jazan region, Saudi Arabia: a cross-sectional study

Maged El-Setouhy[1,2], Ahmad Y. Alqassim[1], Mohammad Zaino[3], Essam A. AlAmeer[4], Anwar Makeen[1], Mohammed A. Muaddi[1], Abdullah A. Alharbi[1], Renad H. Hamzi[1], Amaal A. Hamdi[1], Hanan N. Abu Summah[1], Norah K. Najmi[1], Raghad M. Sharahily[1], Maram A. Zuqayli[1], Fatimah A. Khubrani[1] and Kholod Wasli[5]

[1] Department of Family and Community Medicine, Faculty of Medicine, Jazan University, Jazan, Saudi Arabia
[2] Department of Community, Environmental and Occupational Medicine, Faulty of Medicine, Ain Shams University, Cairo, Egypt
[3] Faculty of Nursing and Health Science, Physical Therapy Department, Jazan University, Jazan University, Jazan, Saudi Arabia
[4] Psychiatry Department, Prince Sultan Military Medical City, Riyadh, Saudi Arabia
[5] AlHokamaa Specialist Center, Jazan, Saudi Arabia

## ABSTRACT

**Background**. Autism spectrum disorder (ASD) is a multifaceted neurodevelopmental condition marked by distinct behavioral patterns, impaired social interaction, and communication challenges. Early recognition and appropriate intervention are pivotal for improving outcomes. This study aims to comprehensively evaluate the understanding, attitudes, and teaching methodologies of kindergarten and primary school teachers in the Jazan region, Kingdom of Saudi Arabia, regarding children with ASD.

**Methodology**. A cross-sectional study was conducted using a multistage cluster random sampling technique. An interview-based questionnaire was administered to kindergarten and primary school teachers in the Jazan region. The target sample size was 800 teachers.

**Results**. Among the 870 participating teachers, 87.8% reported lacking prior training on effectively addressing the needs of children with ASD. However, 74.8% demonstrated substantial understanding of the social communication difficulties faced by autistic children, and 76.2% were aware of the attention-related challenges these children encounter. Additionally, 77.7% of participants recognized the pivotal role of open communication between teachers and families in facilitating quality educational experiences and enhancing academic outcomes for students with ASD.

**Conclusion**. The study highlights the critical need for targeted training programs to equip teachers with essential skills for supporting students with ASD. These findings underscore the importance of policy interventions to ensure adequate resources and expertise for effectively accommodating the unique needs of students with ASD in mainstream schools.

Corresponding author
Maged El-Setouhy,
maged.a.elsetouhy@gmail.com

## INTRODUCTION

Autism spectrum disorder (ASD) is a neurodevelopmental condition that typically manifests by the age of two years, profoundly affecting an individual's behavior, social interaction, communication skills, and learning abilities (*National Institute of Mental Health (NIMH), 2024*; *Saudi Ministry of Health (MOH), 2021*). This intricate condition is believed to be influenced by a combination of genetic and environmental factors (*World Health Organization (WHO), 2023*). While significant advancements have been made in improving the symptoms and functioning of individuals with ASD, achieving positive outcomes necessitates early diagnosis and targeted interventions (*Helt et al., 2008*; *Posar & Visconti, 2019*).

In this context, kindergarten and primary school teachers emerge as pivotal figures. Their familiarity with ASD not only aids in early identification but also supports parents in navigating the diagnostic and intervention landscape (*Khan et al., 2020*). Globally, the World Health Organization (WHO) estimated that one in every 100 children experiences ASD (*World Health Organization (WHO), 2023*). On the other hand, Centers of Disease control and prevention declared that ASD increased in the United States (US) among 8-year-old children from 1/150 in 2000 to 1/36 in 2018 (*Maenner et al., 2023*). This prevalence of ASD is escalating, with males being affected more frequently than females at a ratio of 4.5 to 1 (*Khan et al., 2020*). Arab Gulf countries, including Saudi Arabia, witnessed an increase in ASD incidence from 1.4 to 29 per 10,000 people in 2014 (*Salhia et al., 2014*). In Saudi Arabia, a study in the western region reported an ASD prevalence of 2.81 per 1,000 children, as estimated from the numbers treated in the ASD centers. However, the actual prevalence is believed to be around 42,500, reflecting a challenge attributed to the limited number of ASD centers in the country (*Sabbagh et al., 2021*).

Early identification of ASD crucial for facilitating timely interventions, which can significantly improve developmental outcomes for children diagnosed with ASD (*Gabbay-Dizdar et al., 2022*; *Okoye et al., 2023*). Research underscored the significance of early diagnosis, with studies affirming that identifying and treating ASD early can lead to the resolution of developmental challenges by adulthood (*Anderson, Liang & Lord, 2014*; *Bajko & Bazgan, 2017*; *Steinhausen, Mohr Jensen & Lauritsen, 2016*).

Despite the potential benefits of early intervention, research reveals a concerning gap in ASD knowledge among teachers, impacting the chances of timely intervention (*Al-Hendawi et al., 2023*; *Darling-Hammond & Berry, 2006*; *Fleury et al., 2014*). Furthermore, the academic achievement of autistic children is intricately linked to communication, social, and behavioral difficulties, presenting formidable obstacles in school settings (*Grimm et al., 2018*).

In some areas of the world studies on teacher awareness of ASD have demonstrated varying levels of knowledge. In India, 95.7% of teachers were aware of ASD, while awareness levels were lower in Nigeria (66%) and Pakistan (71.2%), indicating a need for improvement in some regions. These findings underscore a trend of inadequate awareness among teachers regarding the needs of children with ASD and effective strategies for addressing those needs

(*Ayub et al., 2017*; *Paul & Gabriel-Brisibe, 2015*; *Shetty & Rai, 2014*; *Paul & Gabriel-Brisibe, 2015*; *Shetty & Rai, 2014*).

In Saudi Arabia, studies conducted in the Al-Qassim (central) and Jeddah (west) regions reported ASD knowledge levels among teachers at 48.7% and 58%, respectively. These studies emphasized the substantial influence of teachers' attitudes, education levels, and interactions with children with autism on their understanding of ASD (*Alharbi et al., 2019*; *Khalil et al., 2020*). However, the southern region, particularly Jazan, which is also a frontier region adjacent to Yemen, remains unexplored in this context, rendering this study unique in its contribution.

Given the pivotal role teachers play in shaping a child's educational journey, understanding their knowledge, attitude, and practices regarding ASD is paramount. A recent master thesis in the United States conducted a systematic review emphasized the need for better teachers' preparedness in supporting autistic children (*Gallardo, 2024*). While teachers generally have positive attitude towards the inclusion of autistic children in regular schools; further research and extensive training are necessary to enhance their ability to enhance their ability to effectively accommodate autistic students (*Russell, Scriney & Smyth, 2023*).

This study aimed to fill the existing gap by assessing teachers' knowledge of early ASD signs, identifying common misconceptions, evaluating attitudes toward children with ASD, and understanding the methods employed by teachers to address the needs of this population in the Jazan region, Saudi Arabia. Through this exploration, the study aspired to shed light on the state of teacher preparedness concerning ASD in a region where this aspect has not been comprehensively studied, offering valuable insights for future educational strategies and interventions.

## MATERIALS & METHODS

### Study design, sitting, period, and sampling technique

In Saudi Arabia, governmental schools are gender-segregated from the first grade, with boys and girls attending separate schools. Children diagnosed with ASD are typically enrolled in special education schools. However, our study focuses on mainstream governmental schools, where autistic children may remain undiagnosed and unidentified.

We conducted a cross-sectional study in the Jazan Region, situated in the far southwest of the Kingdom of Saudi Arabia, spanning the period from January to April 2023. Jazan region is one of the 13 regions of Saudi Arabia. Jazan region has a homogenous population structure. Nevertheless, we employed a rigorous multistage cluster random sampling to ensure a comprehensive and representative sample.

In the initial phase, five governorates—namely, Jazan, Abu Arish, Samta, Al-Ardah, and Al-Ahad—were randomly selected from the 16 governorates comprising the region. Subsequently, a second stage involved the random selection of four male and four female schools from both kindergarten and primary schools within each chosen governorate. Finally, in the third stage, we invited all available teachers in each school to participate in the study.

This comprehensive approach ensures a representative sample by strategically selecting clusters and then employing random sampling within those clusters. Moreover, the inclusion of both genders and various age groups from kindergarten and primary schools enhances the diversity of the sample.

## Sample size calculation

We used the Raosoft sample size calculator to calculate our sample size (*Raosoft, 2004*). With a total population of approximately 20,000 teachers in kindergarten and primary schools within the Jazan region, a 3% margin of error, and a 90% confidence level, the calculated sample size stood at 725. In anticipation of potential nonresponses, we increased the minimal required sample size to be 800. This proactive approach enhances the study's reliability and ensures that the findings are both statistically sound and resilient (*Toepoel & Schonlau, 2017*).

## Participants

We visited the randomly chosen schools, interviewed all teachers, and explained the purpose of the research. Those who agreed to participate in the study completed our anonymous interview questionnaire after verbally obtaining informed consent.

## Study tools

We used an interview-based questionnaire crafted in Arabic in Google Forms. This comprehensive tool delved into various aspects, including teachers' knowledge of early ASD signs, their perceptions of common misconceptions, attitudes toward children with ASD, and the strategies they used to employ to meet the needs of this specific population. The initial set of questions was adapted from a prior study conducted in 2017 (*Alyami et al., 2022*) and underwent meticulous modifications to align with the characteristics of our target population.

We developed the questionnaire in four sections. The first section was related to the demographic data of the participant teachers, including their age, gender, marital status, having children, living in a city or urban area, highest degree, and getting scholarships or not. The second section was designed to collect general information about autism including their knowledge about it, where did they get their knowledge, having an autistic child or a relative with ASD, how they consider ASD and its relation to family history, vaccination, etc. The third section was collecting data about the early signs that would appear in reference to the autistic children. The fourth section was designed to find what approaches the teachers used when teaching children with ASD.

We added four questions that were intended to add a layer of depth to the understanding of teachers' knowledge about ASD. By exploring aspects that are subject to ongoing discussions, the study not only assessed the current awareness of teachers but also acknowledged the dynamic and evolving nature of our understanding of ASD. This approach provided a nuanced perspective on the challenges in establishing consensus on certain aspects of ASD within the educational context, contributing to a more comprehensive evaluation of teachers' knowledge in this critical area.

In order to assess the practicality and efficiency of the interview-based approach, we conducted a pilot study involving 30 participants (results not included). The primary objectives of the pilot study were to evaluate the relevance of the questions and determine the time required for efficient data collection. Drawing insights from the pilot study, we made the necessary adjustments to refine the questionnaire, ensuring clarity and pertinence.

## Statistical analysis

The collected data underwent rigorous analysis utilizing the Statistical Product and Service Solutions (SPSS) software version 27 (IBM Corp., Armonk, NY, USA). Descriptive statistics, encompassing mean, standard deviation, frequencies, and percentages, were employed to provide a comprehensive overview of both quantitative and categorical variables. In order to discern variations among participants, the $T$-test was used to compare differences between two categories of independent variables. For independent variables with more than two categories, a one-way analysis of variance (ANOVA) was conducted to assess differences among the groups. Following a significant ANOVA result, *post hoc* pairwise comparisons were performed using the Bonferroni correction to control for multiple comparisons and minimize the risk of Type I errors. This approach ensures that the observed differences between groups are statistically rigorous and reliable. The predetermined significance level was set at $P < 0.05$, with results reported alongside a 95% confidence interval (CI) to underscore both statistical significance and precision.

## Ethical consideration

Prior to initiation, the study received thorough scrutiny and ethical approval from the Research Ethics Committee (REC) at Jazan University, denoted by reference number REC-44/06/460. This ethical endorsement ensured the study's alignment with ethical guidelines and principles. Informed consent forms, meticulously outlining the study's purpose and procedures, were provided to participants. Participants demonstrated their voluntary agreement by reading, comprehending, and verbally accepting the terms stipulated in the informed consent forms.

Participants were explicitly informed of their right to abstain from participation or withdraw from the study at any juncture, emphasizing the principle of voluntary participation. To safeguard participant confidentiality and privacy, stringent measures were implemented. The collected data were anonymized, securely stored, and access was restricted to the research team, reinforcing the commitment to upholding the welfare and rights of study participants throughout the entire research endeavor. All data will be stored in a password secured file on the principal investigator computer for at least 5 years as requested by the ethical committee.

## RESULTS

Table 1 provides a snapshot of the demographic characteristics of the 870 teachers who participated in this study. Their average age was 42.3 years ($\pm$ 6.9 SD), and their collective teaching experience averaged 16 years ($\pm$ 7.6 SD). A significant majority (78.6%) of teachers were familiar with the term "Autism". Regarding their sources of knowledge on

**Table 1** Demographic characteristics and knowledge about ASD among Jazan school teachers.

| Variables (Respondents overall 870) | N (%) |
|---|---|
| Gender (Respondents 870) | |
| Male | 418 (48) |
| Female | 452 (52) |
| Marital status (Respondents 868) | |
| Single | 731 (84) |
| Married | 85 (9.8) |
| Widow | 15 (1.7) |
| Divorced | 37 (4.3) |
| Children (Respondents 862) | |
| Has children | 712 (81.8) |
| Does not have children | 150 (17.4) |
| Place of living (Respondents 857) | |
| Urban | 355 (41.1) |
| Rural | 502 (58.6) |
| Highest degree (Respondents 865) | |
| Diploma | 148 (17) |
| Bachelor's | 680 (78.6) |
| Master's | 36 (4.2) |
| Ph.D. | 1 (0.1) |
| Knowing the term "ASD" | 682 (78.4) |
| The source of information about ASD (more than one answer was allowed) | |
| Television | 243 (27.9) |
| Friends | 222 (25.5) |
| Relatives | 151 (17.4) |
| Books and articles | 187 (21.5) |
| Seminars or lectures | 100 (11.5) |
| Social media | 363 (41.7) |
| Autistic person | 91 (10.5) |
| Others | 8 (0.9) |
| Has an autistic child | 13 (1.5) |
| Has an autistic relative | 241 (27.7) |
| Had contact with an autistic child | 272 (31.3) |

ASD, 27.9% mentioned television, 25.5% cited friends, and the highest percentage (41.7%) identified social media as their primary source of information on ASD.

Table 2 presents findings on teachers' general knowledge about ASD and recognition of its early signs in autistic children. More than half of the participants (51.5%) correctly acknowledged that autistic children are not mentally disabled. Regarding perceived risk factors, 71.7% associated electronic devices with ASD, while 14.4% linked early-age vaccination to ASD. Furthermore, 37.1% considered the way the family raise their children as a potential influencing factor, and 59.7% believed that autistic children can be cured.

**Table 2  Jazan teacher's knowledge about ASD.**

| Variables | Yes N (%) | No N (%) | I don't know N (%) | Total N (%) |
|---|---|---|---|---|
| Knowledge | | | | |
| ASD is an organic disease | 128 (14.7) | 406 (46.7) | 336 (38.6) | 870 (100) |
| ASD children are mentally disabled | 168 (19.3) | 448 (51.5) | 254 (29.2) | 870 (100) |
| Hyperbaric oxygen can treat ASD | 90 (10.3) | 98 (11.3) | 551 (63.3) | 739[*] (100) |
| Diet can play a role in treating ASD | 151 (17.4) | 123 (14.1) | 448 (51.5) | 722[**] (100) |
| Genetic predisposition is a risk factor for ASD | 409 (47) | 198 (22.8) | 263 (30.2) | 870 (100) |
| Electronic devices are a risk factor for ASD | 624 (71.7) | 109 (12.5) | 137 (15.7) | 870 (100) |
| Early vaccination is a risk factor for ASD | 125 (14.4) | 407 (46.8) | 338 (38.9) | 870 (100) |
| How family treat the child is a risk factor for ASD | 323 (37.1) | 304 (34.9) | 243 (27.9) | 870 (100) |
| ASD is curable | 519 (59.7) | 138 (15.9) | 213 (24.5) | 870 (100) |
| Early signs of ASD | | | | |
| Lack of communication | 651 (74.8) | 109 (12.5) | 110 (12.6) | 870 (100) |
| Disturbed language | 589 (67.7) | 98 (11.3) | 183 (21.0) | 870 (100) |
| Inability to express their feelings | 429 (49.3) | 194 (22.3) | 247 (28.4) | 870 (100) |
| Involuntary movements or behaviors | 573 (65.9) | 57 (6.6) | 240 (27.6) | 870 (100) |
| Eating disorders | 380 (43.7) | 108 (12.4) | 382 (43.9) | 870 (100) |
| Sensory issues | 367 (42.2) | 113 (13.0) | 390 (44.8) | 870 (100) |
| Hyperactivity | 550 (63.2) | 129 (14.8) | 191 (22.0) | 870 (100) |
| Attention deficit | 663 (76.2) | 37 (4.3) | 170 (19.5) | 870 (100) |

Notes.
[*]The number of non-responders to the question was 131 (15.1).
[**]The number of non-responders to the question was 148 (17).

Exploring these additional points provide deeper insights into teachers' beliefs, enriching the overall findings of the study.

In the second part of Table 2, we presented a detailed examination of teachers' awareness regarding early signs of ASD. All indicators listed in this section were acknowledged by teachers as significant early signs of ASD. Notably, a substantial percentage of teachers demonstrated a high level of awareness, with 76.2% recognized attention deficit as a crucial

**Table 3  Some beliefs of the Jazan kindergarten and primary school teachers about ASD and autistic children.**

| Variable | Always N (%) | Sometimes N (%) | Rarely N (%) | Never N (%) | Don't know N (%) | Total N (%) |
|---|---|---|---|---|---|---|
| Autistic children have higher skills | 139 (16) | 428 (49.2) | 104 (12) | 30 (3.4) | 169 (19.4) | 870 (100) |
| Autistic children have name response deficit | 123 (14.1) | 441 (50.7) | 93 (10.7) | 44 (5.1) | 169 (19.4) | 870 (100) |
| Regular school's environment has sufficient knowledge about ASD | 45 (5.2) | 208 (23.9) | 254 (29.2) | 222 (25.5) | 141 (16.2) | 870 (100) |
| Regular school's environment is appropriate to meet the needs of Autistic children | 33 (3.8) | 178 (20.5) | 164 (18.9) | 370 (42.5) | 125 (14.4) | 870 (100) |

early sign, and 74.8% acknowledged the importance of communication deficits. However, there were variations in awareness levels, with 49.3% recognized the challenges autistic children face in expressing their feelings, and 42.2% recognized the sensory issues.

Table 3 demonstrates some believes of the Jazan teachers regarding ASD. More than half of the teachers (65.2%) believed that autistic children have higher skills than their peers although most of them (64.8%) believed that those children have name response deficit as a sign of ASD. On the other hand, half of the teachers believed that regular school environment is not suitable for autistic children.

Regarding the teachers' abilities and methods they use when teaching autistic children, such as creating an educational plan and using suitable methods for their abilities, we discovered that 87.8% of the teachers did not receive any prior training on how to teach autistic children. Moreover, 72.3% of the teachers believed that regular schools do not provide activities that are in line with the abilities of children with ASD. In addition, more than half of the participants (59.1%) indicated that regular schools in the region lack specialized assistants to deal with ASD children (results not presented in tables).

We employed a one-way ANOVA analysis as well as the $T$-test to compare the effect of each of the variables; gender, marital status, having a child with ASD, and having a relative with ASD, and latest qualification of the teachers on three components; knowledge about ASD, knowledge about early signs of ASD, ability and methods in dealing with autistic children (Table 4).

The scores of each component were created as the sum of the items constructing that component. Higher scores referred to more knowledge or more ability in dealing with autistic children.

The result displayed in Table 4 revealed that single teachers have more knowledge about ASD compared to other teachers; However the difference was statistically not significant ($P = 0.222$). Moreover, the knowledge about ASD was higher among teachers with higher educational levels ($P < 0.001$). Exploring these additional points provide deeper insights into teachers' beliefs, enriching the overall findings of the study.

Furthermore, the results show that male teachers' level of knowledge about early signs of ASD was 15.80 on average with SD = 3.5. The female average scores were 15.2 with SD = 3.7. Similarly, for the component ''teachers' ability in dealing with autistic children''.

**Table 4  Univariate analysis for testing the significance differences in teachers' knowledge about ASD and its early signs and ability in dealing with ASD in relation to demographic and personal characteristics.**

| Variable | Category | N | Mean | 95% CI for mean | P-value | Post hoc P-value |
|---|---|---|---|---|---|---|
| **Jazan teachers' knowledge about ASD** | | | | | | |
| Gender | Male | 414 | 16.6 ± 3.8 | [16.3, 17.0] | 0.090$^T$ | – |
| | Female | 450 | 16.2 ± 4.2 | [15.8, 16.6] | | – |
| Marital status | Single | 725 | 16.5 ± 4 | [16.2, 16.8] | | Ref.Cat |
| | Married | 85 | 15.6 ± 4.1 | [14.7, 16.5] | 0.222$^A$ | – |
| | Widowed | 15 | 16.3 ± 4.9 | [13.6, 19.0] | | – |
| | Divorced | 37 | 16.1 ± 4.2 | [14.6, 17.5] | | – |
| Has a child with ASD | Yes | 13 | 19.2 ± 3 | [17.4, 20.9] | 0.130$^T$ | – |
| | No | 851 | 16.4 ± 4 | [16.1, 16.6] | | – |
| Has a relative with ASD | Yes | 239 | 18 ± 2.9 | [17.7, 18.4] | 0.000$^T$ | – |
| | No | 625 | 15.8 ± 4.2 | [15.5, 16.1] | | – |
| Educational level | Diploma | 148 | 15.4 ± 4.3 | [14.7, 16.1] | | Ref.Cat |
| | Bachelor's | 680 | 16.5 ± 3.9 | [16.2, 16.8] | 0.000$^A$ | 0.000 |
| | Master's | 36 | 18.1 ± 3.1 | [17.0, 19.1] | | 0.001 |
| **Jazan teachers' knowledge about early signs of ASD** | | | | | | |
| Gender | Male | 345 | 15.8 ± 3.5 | [15.4, 16.2] | 0.023$^T$ | – |
| | Female | 369 | 15.2 ± 3.7 | [14.8, 15.6] | | – |
| Marital status | Single | 599 | 15.6 ± 3.6 | [15.3, 15.9] | | Ref.Cat |
| | Married | 70 | 15.6 ± 3.7 | [14.7, 16.5] | 0.018$^A$ | 1.000 |
| | Widowed | 14 | 13.8 ± 2.9 | [12.1, 15.5] | | 0.177 |
| | Divorced | 30 | 13.8 ± 3.6 | [12.5, 15.2] | | 0.027 |
| Has a child with ASD | yes | 11 | 15.7 ± 2.7 | [13.9, 17.5] | 0.823$^T$ | – |
| | no | 703 | 15.5 ± 3.6 | [15.2, 15.7] | | – |
| Has a relative with ASD | yes | 197 | 16.2 ± 30 | [15.8, 16.7] | 0.001$^T$ | – |
| | no | 517 | 15.2 ± 3.8 | [14.9, 15.5] | | – |
| Educational level | Diploma | 128 | 14.8 ± 3.8 | [14.1, 15.5] | | Ref.Cat |
| | Bachelor's | 562 | 15.6 ± 3.5 | [15.3, 15.9] | 0.000$^A$ | 0.045 |
| | Master's | 24 | 16.8 ± 3.3 | [15.4, 18.2] | | 0.020 |
| **Jazan teachers' ability in dealing with ASD Children** | | | | | | |
| Gender | Male | 414 | 13.0 ± 2.3 | [12.8,13.2] | 0.006$^T$ | – |
| | Female | 450 | 12.6 ± 2.5 | [12.3,12.8] | | – |
| Marital status | Single | 725 | 12.9 ± 2.3 | [12.7,13.0] | | Ref.Cat |
| | Married | 85 | 12.5 ± 2.5 | [11.9,13.0] | 0.190$^A$ | |
| | Widowed | 15 | 13 ± 2.8 | [11.4,14.6] | | |
| | Divorced | 37 | 12.2 ± 2.9 | [11.2,13.1] | | |
| Has a child with ASD | yes | 13 | 13.5 ± 1.3 | [12.8,14.3] | 0.254$^T$ | – |
| | no | 851 | 12.8 ± 2.4 | [12.6,12.9] | | – |

| Variable | Category | N | Mean | 95% CI for mean | P-value | Post hoc P-value |
|---|---|---|---|---|---|---|
| Has a relative with ASD | yes | 239 | 13.3 ± 1.9 | [13.0,13.5] | 0.000[T] | – |
|  | no | 625 | 12.6 ± 2.5 | [12.4,12.8] |  | – |
| Educational level | Diploma | 148 | 12.3 ± 2.7 | [11.9,12.8] |  | Ref.Cat |
|  | Bachelor's | 680 | 12.9 ± 2.3 | [12.7,13.0] | 0.025[A] | 0.023 |
|  | Master's | 36 | 13.2 ± 1.9 | [12.5,13.8] |  | 0.093 |

**Notes.**

For all variables with two categories T-Test is used and its P-value is denoted by the subscript T. When more than two categories ANOVA (Analysis of Variance) is conducted with overall P-value denoted by subscript A. The 95% CI: refers to 95% confidence interval for the man value. Ref.Cat: Reference category that used by Bonferroni test to do a comparison with other categories. *Post hoc* P-value: (*post hoc*, Dunnet *t*-test) calculated only in case of overall P-value were significant and the variable has 3 or more categories. These values are showing the statistically significant difference between the reference category and each other category.

The male teachers indicated increasing ability compared to the female teachers ($P = 0.006$, 95% CI [12.8, 13.2]).

The results in Table 4 show that a single teacher is more likely to have more knowledge about early signs of ASD compared to a divorced teacher ($P = 0.027$).

In addition, among those teachers who having a relative with ASD the result revealed that they have significantly more knowledge about ASD, about early signs of ASD and better ability to dealing with autistic children compared to teachers without having relatives with ASD ($P < 0.001$, 95% C.I [17.7,18.4], $P = 0.001$, 95% C.I. [15.8,16.7], $P < 0.001$, 95% C.I [13.0,13.5], respectively).

Moreover, it is found that, the higher the education levels of the teachers the more his/her knowledge about ASD and more knowledge about early sign ASD and better ability in dealing with autistic children ($P < 0.001$, $P < 001$, $P = 0.025$ respectively).

# DISCUSSION

Our study shed light on significant deficiencies in the general understanding of ASD among kindergarten and primary school teachers, particularly in the Jazan region in the far southwest of Saudi Arabia. Although the prevalence of ASD is alarming all over the world and increasing than expected in some areas in the world who have good statistics for the disease (*Maenner et al., 2023*), only a small fraction of the teachers in Jazan (14.7%) recognized ASD as an organic neurodevelopmental disease (*Kim, 2015*), with a substantial percentage (38.6%) demonstrated unclear comprehension of the topic. Furthermore, a concerning misconception emerged, with nearly one-fifth (19.3%) of teachers linking autism with mental retardation, although it is not (*Ghosh & Gorakshakar, 2009*; *Melvin et al., 2022*). These findings underscore a widespread lack of accurate knowledge and misconceptions surrounding ASD among educators in Jazan (Table 2).

This pattern of insufficient awareness among teachers mirrors similar studies conducted across the Kingdom of Saudi Arabia. For instance, a 2019 study in the Al-Qassim region highlighted comparable deficits in teacher understanding of ASD, suggesting a systemic issue transcending regional boundaries which did not change between 2013 and 2022 (*Alharbi et al., 2019*; *Alobaid & Almogbel, 2022*). Recognizing these knowledge gaps, we

sought to enhance the robustness of our study by increasing the sample size, motivated by the imperative to address these pervasive misconceptions.

In our investigation into teachers' perceptions of ASD risk factors, a significant majority (71.7%) identified electronic devices as potential contributors to ASD development (Table 2). This finding resonates with a study conducted in Saudi Arabia, where more than half of the participants (55.4%) supported the idea that limiting electronic device usage among children could aid in treating or preventing ASD (*Khalifa et al., 2023*). This alignment underscores the widespread concern regarding electronic device usage as a perceived risk factor for ASD among educators in Saudi Arabia (*Alrahili et al., 2021*).

Some beliefs are even dangerous like the fake belief linking child vaccination with the development of ASD (*Baird et al., 2008*; *Gabis et al., 2022*; *Taresh et al., 2020*). On comparing our results to an Australian study, our own showed a concerning disparity in beliefs regarding child vaccination and its association with ASD. While the Australian study showed a 3% agreement with the notion of vaccines as a risk factor for ASD, our study revealed a significantly higher percentage (14.4%) of participants attributing ASD development to vaccination, with an additional 38% expressing uncertainty on this matter (Table 2). This discrepancy underscores the urgent need for comprehensive education and awareness initiatives, particularly targeting teachers who play a pivotal role in shaping community perceptions and understanding of ASD risk factors (*Jones et al., 2021*).

On the other hand, although diet can play a role in managing ASD (*Alam, Westmark & McCullagh, 2023*; *Hartman & Patel, 2020*), most of our teachers (51.5) did not know that. This goes in agreement with another study conducted in KSA in 2023, where a high percentage of participants (47.3) did not know anything about using diet for managing ASD (*Alam, Westmark & McCullagh, 2023*). Furthermore, our findings indicated a troubling misconception among a notable portion (37.1%) of teachers who erroneously believe that family interactions may predispose them to ASD (Table 2). While family dynamics undoubtedly influence child development, it is crucial to clarify that ASD is a multifaceted condition with genetic, environmental, and neurological components. Educating stakeholders about the complex nature of ASD and dispelling misconceptions about its etiology is imperative for fostering a more informed and supportive environment for individuals and families affected by ASD (*Jones et al., 2021*). In addition, around 60% of the surveyed teachers expressed a belief in the possibility of curing ASD (Table 2). However, it is crucial to understand that ASD is typically a lifelong condition, although symptoms can be effectively managed and improved with early appropriate interventions. While recovery from ASD is not impossible, it is essential to clarify that it does not guarantee a complete cure of the condition as researchers indicated that up to 25% of individuals with ASD may no longer meet the diagnostic criteria after a certain period. Nevertheless, even if they no longer meet the criteria, they may still exhibit some autistic traits, albeit to a lesser degree (*Helt et al., 2008*; *Posar & Visconti, 2019*; *Whiteley, Carr & Shattock, 2019*). It is important to emphasize that recovery from ASD does not equate to attaining perfect mental and social health similar to neurotypical individuals. Regarding the genetic predisposition as a risk factor for ASD, nearly half (47%) of our teachers believed that it is a risk factor for developing ASD. Likewise, an Australian study showed that 45.8% of those who have no

relation to ASD believed that the condition is inherited with even increase in the percentage of those who agreed for that among the parents of the autistics (65.5%) and those who had a closed autistic relatives (56.7%) (*Jones et al., 2021*).

In the second part of our study, we delved into the knowledge of kindergarten and primary school teachers concerning the primary early signs of ASD, recognizing the pivotal role their understanding plays in facilitating timely referral and intervention management (*Zwaigenbaum et al., 2015*). Our analysis uncovered significant gaps in teachers' awareness within this domain. Of particular concern was the notably low level of knowledge reported regarding sensory problems experienced by autistic children (42.2%). Despite numerous studies underscoring that a majority of autistic children grapple with sensory difficulties, this aspect remained largely overlooked among the surveyed teachers (*Fabbri-Destro et al., 2022*; *McCormick et al., 2016*; *Parmeggiani, Corinaldesi & Posar, 2019*). Similarly, only 43.7% demonstrated awareness of eating disorders in autistic children, despite research indicating a prevalence of approximately 60% in this population (*Parmeggiani, Corinaldesi & Posar, 2019*; *Parsons, 2023*). Another notable finding was the lack of awareness (49.3%) among teachers regarding the struggles autistic children face in expressing their emotions, despite evidence suggesting that around one-third of ASD children exhibit impaired emotional expression (*Molnar-Szakacs et al., 2009*; *Parmeggiani, Corinaldesi & Posar, 2019*). Conversely, our study found that teachers demonstrated the highest levels of awareness regarding attention deficit (76.2%) and lack of communication (74.8%) as early signs of ASD. These findings indicate a relatively stronger understanding in these particular areas among the surveyed teachers. However, it is noteworthy that all other aspects of knowledge concerning the early signs of ASD fell below the 70% mark, indicating significant room for improvement across the board. However, these findings were also demonstrated in a study conducted in Pakistan that reported a considerable proportion of teachers, were aware of poor communication (53%–75.3%) and attention deficit (68%–69%) as early signs of ASD (*Arif et al., 2013*). Other studies in Yemen and Oman. In Yemen, most teachers were unfamiliar with a range of early signs of ASD, including difficulties in social communication and engagement, limited and repetitive activities, and sensory sensitivity (*Taresh et al., 2020*). The results of Oman also showed even higher a lack of knowledge (*Al-Sharbati et al., 2015*). The same results with different levels were reported in 25 articles around the world (*Gómez-Marí, Sanz-Cervera & Tárraga-Mínguez, 2021*). This lack of knowledge would be the main reason that 63.4% of our teachers considered regular schools are not suitable for autistic children. Most of them also reported (59.1%) of the schools lack specialized assistants. However, another study thought that regular schools would be beneficial for the development of autistic children (*Fleury et al., 2014*). On the other hand, a recent Saudi study considered that the inadequate knowledge of the Saudi teachers and their lack of training in this area would hinder this in Saudi schools (*Khalil et al., 2020*). This was clearer to us when we found that more than two thirds (72.3%) of our teachers received no training to deal effectively with autistic children. Several studies in different countries have shown similar results. A study about the needs of students with ASD, their findings showed that the teachers often were not educated enough on how to deal effectively with ASD students (*Boujut et al., 2016*). A qualitative research study was conducted to

address teacher's challenges and preparation needs when dealing with ASD students and the outcomes of this study suggest that existing teacher education programs often do not adequately prepare educators to resolve challenges associated with teaching ASD students (*Boujut et al., 2016*). Our results confirm and extend those findings and suggest that the pre-training of primary and kindergarten teachers is of great importance in dealing with ASD children, as there is no training program concerned with this category, and most of the teachers who obtained this training were from attending training workshops outside the scope of preparing teachers (*Busby et al., 2018*).

Finally, we tried to determine the teachers' demographic characteristics that affected the three main components of the study, namely knowledge about ASD and its early signs and the ability to dealing with autistic children. Not surprisingly, the education level of the teachers was affecting the three components (Table 4). The higher the education levels the higher the scores in all components. This was also the case in different published studies (*Al Jaffal, 2022*; *Alharbi et al., 2019*; *Gómez-Marí, Sanz-Cervera & Tárraga-Mínguez, 2021*).

### Limitations and strengths

While our study provides valuable insights into teachers' understanding, perceptions, and practices regarding ASD in the Jazan region, it is important to recognize certain limitations. Firstly, the cross-sectional design hinders our ability to establish causality and track changes over time. Longitudinal research would offer more robust evidence. Secondly, the study's sample was limited to the Jazan region, potentially affecting the generalizability of findings to regions with different cultural, educational, and healthcare contexts. Nonetheless, our findings contribute significantly to understanding ASD in the Jazan region and may have implications for similar Saudi regions as for the southern region of Saudi Arabia and other frontier areas in the Gulf and Arab countries.

## CONCLUSIONS

Our study identified significant gaps in teachers' understanding of early signs of ASD, highlighting familiarity but lacking comprehensive comprehension. Notably, most teachers lacked specialized training in identifying and supporting autistic students, indicating a critical deficiency in their professional development. This emphasizes the urgent need to address teachers' readiness in this regard, as it affects their ability to effectively support these students. Investing in tailored training initiatives to enhance teachers' understanding of ASD is crucial for creating inclusive learning environments. These findings emphasize the necessity for nationwide educational initiatives and professional development programs aimed at improving teacher awareness and understanding of ASD, like ASD-specific strategies, early signs recognition, and tailored support. Our findings will be helpful for the decision makers to plan with our team to develop a training program for primary school teachers to be prepared for autistic children.

## ACKNOWLEDGEMENTS

We express our sincere appreciation to the administrators and teachers of kindergarten and primary schools in the Jazan region for their unwavering cooperation and enthusiastic

participation in our study. Their invaluable contributions have been indispensable to the successful completion of this research endeavor. We firmly believe that their substantial input will not only enhance the findings of this study but also lay a solid foundation for future research in this critical field.

### Funding
This work received no funds from any funding agent.

### Competing Interests
The authors declare there are no competing interests. We have no financial, non-financial, professional, or personal conflict of interest of any type. Kholod Wasli is employed by AlHokamaa Specialist Center and has no competing interest as she helped us with the points mentioned as an author with no financial support from her work or from the group of researchers.

### Author Contributions
- Maged El-Setouhy conceived and designed the experiments, analyzed the data, prepared figures and/or tables, authored or reviewed drafts of the article, project management, and approved the final draft.
- Ahmad Y. Alqassim conceived and designed the experiments, analyzed the data, prepared figures and/or tables, authored or reviewed drafts of the article, project management, and approved the final draft.
- Mohammad Zaino analyzed the data, prepared figures and/or tables, authored or reviewed drafts of the article, and approved the final draft.
- Essam A. AlAmeer conceived and designed the experiments, prepared figures and/or tables, authored or reviewed drafts of the article, and approved the final draft.
- Anwar Makeen conceived and designed the experiments, prepared figures and/or tables, authored or reviewed drafts of the article, and approved the final draft.
- Mohammed A. Muaddi performed the experiments, analyzed the data, prepared figures and/or tables, authored or reviewed drafts of the article, and approved the final draft.
- Abdullah A. Alharbi performed the experiments, analyzed the data, prepared figures and/or tables, authored or reviewed drafts of the article, and approved the final draft.
- Renad H. Hamzi performed the experiments, prepared figures and/or tables, authored or reviewed drafts of the article, and approved the final draft.
- Amaal A. Hamdi performed the experiments, prepared figures and/or tables, authored or reviewed drafts of the article, and approved the final draft.
- Hanan N. Abu Summah performed the experiments, prepared figures and/or tables, authored or reviewed drafts of the article, and approved the final draft.
- Norah K. Najmi performed the experiments, prepared figures and/or tables, authored or reviewed drafts of the article, and approved the final draft.
- Raghad M. Sharahily performed the experiments, prepared figures and/or tables, authored or reviewed drafts of the article, and approved the final draft.

- Maram A. Zuqayli performed the experiments, prepared figures and/or tables, authored or reviewed drafts of the article, and approved the final draft.
- Fatimah A. Khubrani performed the experiments, prepared figures and/or tables, authored or reviewed drafts of the article, and approved the final draft.
- Kholod Wasli conceived and designed the experiments, performed the experiments, prepared figures and/or tables, authored or reviewed drafts of the article, and approved the final draft.

### Human Ethics

The following information was supplied relating to ethical approvals (i.e., approving body and any reference numbers):

This research was approved by the Jazan University Research Ethics Committee. (REC-44/06/460).

### Data Availability

The raw measurements are available in the Supplementary File.

### Supplemental Information

Supplemental information for this article can be found online at http://dx.doi.org/10.7717/peerj.20044#supplemental-information.

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
