# Peer review of "Teacher preparedness regarding autism spectrum disorder in the Jazan region, Saudi Arabia: a cross-sectional study"

_PeerJ, doi:10.7717/peerj.20044_

## Round 0.1 · original submission · Major Revisions

We apologize for the delay, however we now have 2 reviews for your paper. Both reports are clear as to what is needed, and we invite you to submit a detailed revision.

·

Basic reporting

Not applicable

Experimental design

Not applicable

Validity of the findings

Not applicable

·

Basic reporting

Thank you very much for letting me review this fascinating manuscript. Learning about education in Saudi Arabia is important on an international basis.
Notwithstanding, there are some concerns that I recommend addressing:
Basic Reporting
Lines 45 – 46: "…contributing to the recovery of affected individuals". This sentence is problematic. First – because ASD is not a disease so there is no "recovery", and second because the terminology "affected individuals" is not appropriate.
Line 50 is not accurate. The updated prevalence according to the CDC: About 1 in 36 children has been identified with autism spectrum disorder (ASD) according to estimates from CDC's Autism and Developmental Disabilities Monitoring (ADDM) Network
Line 59: the term "affected children" is inappropriate. Maybe write - …..thereby enhancing developmental outcomes for children diagnosed with ASD.
Line 62: this statement is unclear: "….can lead to the resolution of developmental
challenges by age 19". Do the citated references state that specifically? And even if so – this general statement that early diagnosis can lead to the resolution of developmental challenges is misleading and inaccurate.
Lines 68 – 70: You claim that there is a global trend of inadequate awareness among teachers – but give only three examples – India, Nigeria and Pakistan. This does not make for a global claim. Either add examples globally – or narrow your claim to a specific global region.

Experimental design

Participants – I recommend elaborating and explaining the Saudi Arabian school system. The authors describe selecting four male and four female schools – this should be addressed. Is this gender separation practiced from pre-school? The first grade of elementary school? Also – are there special education schools for children with special needs and those diagnosed with ASD? Or do all children learn in regular, general education schools? All this information is paramount to understanding the sampling process.
Line 103 – what is the meaning of "all available teachers"?
Study tools: It would help the reader if an example of a question would be given for each category.

Validity of the findings

Line 170 states that "More than half of the participants (51.5%) correctly acknowledged that
170 autistic children are not mentally disabled". Notwithstanding, there are indeed children with ASD who have co occurring intellectual disabilities.
Lines 175 – 177 explain the importance of including specific points in the questionnaire. This should not be included in the results section, but rather in the study tools.
Line 197 – I recommend not using the term "dealing with" but rather – teaching.
The same comment is relevant for lines 183 – 190.
Discussion
Line 238 – fix the word shaded – to shed.
Line 242 – the expression "ASD as an organic disease" is not acceptable.
Lines 245 – the relation and co occurrence of ASD and ID is not explained or dealt with accurately throughout this manuscript.
Line 276: the claim that diet can play a role in managing ASD is misleading and has not been proven scientifically.

---

## Round 0.2 · Major Revisions

As you can see, Reviewer 2 still has some concerns which you must address.

·

Basic reporting

Good

Experimental design

NA

Validity of the findings

Good

Additional comments

Accept the manuscript

·

Basic reporting

Thank you for implementing most of the recommended revisions.
There are still some concerns that need to be addressed:
Line 37 - The term affected individuals is not appropriate. Please change it.
Line 59 has an editing typo.
Line 62: Since not all cases lead to the resolution of developmental challenges, even with early intervention and identification, I recommend writing, "Research underscored the significance of early diagnosis, with studies affirming that identifying and treating ASD early can lead in some instances to the resolution of developmental challenges by adulthood "
Lines 77– 79 that were included are important and need some elaboration and editing since the English is incorrect. I would also move them to the added lines 88-91—they are a better fit there. Line 91: I recommend (again) not using the words " deal with." They have a very negative connotation, and it is not appropriate.
The new section is good – but has to be edited – the English is unclear.
The governmental schools in Saudi Arabia are separated into male and female schools from the first primary. The diagnosed autistic children are usually sent to specialized schools for studying. Our study is considering the governmental schools where undiagnosed autistic children are.
What is first primary?
I recommend writing: Children who have a diagnosis of ASD learn in special education schools.
And the last sentence also has to be rewritten – the English is unclear.
Line 110 – needs editing
Line 132: needs to be – those who agreed. Not accepted
The new section about the questionnaire is very good. Parts of it need editing.

Experimental design

No comments

Validity of the findings

The relation and co-occurrence of ASD and ID are not explained or dealt with accurately throughout this manuscript. The response that it will be taken care of in a different manuscript is very problematic.

The claim that diet can play a role in managing ASD is misleading and has not been proven scientifically.
The response that the dietary approach is still under trial underscores the problematic stance. therefore, it is critical to stress this point and not leave it as a claim that diet can play a role in managing autism.

---

## Round 0.3 · Minor Revisions

Please address the remaining reviewer comments.

**Language Note:** The review process has identified that the English language must be improved. PeerJ can provide language editing services - please contact us at [email protected] for pricing (be sure to provide your manuscript number and title). Alternatively, you should make your own arrangements to improve the language quality and provide details in your response letter. – PeerJ Staff

·

Basic reporting

There are still some issues I do not agree with, and I ask the authors to address these issues in the limitations section:
- The relation and co-occurrence of ASD and ID are not explained or dealt with accurately throughout this manuscript. The response that it will be taken care of in a different manuscript is very problematic.
The authors can add in the limitations section that the purpose of this manuscript is mainly to discover the preparedness of the teachers in ordinary schools to deal with ASD students, whether they have ASD alone or with ID, as most of the ASD children have both.

The claim that diet can play a role in managing ASD is misleading and has not been proven scientifically.
The response that the dietary approach is still under trial underscores the problematic stance. Therefore, in the limitations section, the authors can write this.

Other than that, the paper is much improved. There are still many editing issues. I have edited some, but I strongly recommend that an English-speaking editor edit the manuscript.

Experimental design

No comment

Validity of the findings

No comment

---

## Round 0.4 · Minor Revisions

Please address the remaining comments frmo Reviewer 2

·

Basic reporting

A significant improvement in the manuscript. Please edit all tables before publishing them - for example: Title of Table 3: instead of the word "believes," you need to write 'beliefs."
In table 4 - instead of "have a child / relative with autism," it needs to be "has"

Experimental design

no comment

Validity of the findings

no comment

Additional comments

no comment
I think the manuscript is ready for acceptance - it just needs the tables edited.

---

## Round 0.5 · Minor Revisions

Please address the remaining item from Reviewer 2

·

Basic reporting

no comments

Experimental design

no comments

Validity of the findings

no comments

Additional comments

The corrections to tables 3 and 4 are good
One last correction is needed in Table 1.
Instead of writing
Has a child suffering from ASD
Has a relative suffering from ASD
Had contact with a child suffering from ASD
please write
Has a child with ASD
Has a relative with ASD
Had contact with a child with ASD
or
Has an autistic child
Has an autistic relative
Had contact with an autistic child

---

## Round 0.6 · accepted · Accept

Thank you for addressing the reviewer feedback, your article is now accepted for publication.

·

Basic reporting

-

Experimental design

-

Validity of the findings

-